# What Affects Treatment Underuse in Multiple Myeloma in the United States: A Qualitative Study

**DOI:** 10.3390/cancers15082369

**Published:** 2023-04-19

**Authors:** Rose Cytryn, Nina Bickell, Radhi Yagnik, Sundar Jagannath, Jenny J. Lin

**Affiliations:** 1Department of Biomedical Graduate Education, Georgetown University, 3900 Reservoir Road NW, Washington, DC 20057, USA; 2Icahn School of Medicine at Mount Sinai, One Gustave L. Levy Place, Box 1087, New York, NY 10029, USA

**Keywords:** multiple myeloma, treatment underuse, treatment decision making, socioeconomic factors

## Abstract

**Simple Summary:**

Multiple myeloma is the second most common hematologic malignancy. Diagnosis, treatment, and survival rates vary by race, which suggests that race may influence access to or reception of treatment. This study sought to understand multiple myeloma patients’ treatment experiences and factors that influence treatment decision making, barriers to and facilitators of treatment. Through patient interviews, we identified and classified treatment delays or underuse. We found major themes leading to patient decisions, focusing on factors contributing to patients’ delaying or forgoing treatment. Addressing such factors may improve relationships between patients and physicians and patients’ understanding of and access to their care.

**Abstract:**

Background: Multiple myeloma (MM) is the second most common hematologic malignancy. African Americans are more likely than Whites to be diagnosed with and die of MM, but they experience the same survival times in clinical trials, suggesting that differences in survival may be attributed to differences in receipt of treatment or differences in access to new treatments. We undertook this study to identify the reasons and needs underlying disparities in treatment among patients diagnosed with MM. Methods: We conducted in-depth interviews in 2019–2020 with patients diagnosed with MM between 2010 and 2014 who were identified as eligible for transplant and maintenance therapy and having experienced delays in or underuse of treatment for MM. Underuse was defined as the lack of a particular treatment that the patient was eligible to receive, not being transplanted if eligible, and/or not receiving maintenance therapy. Underuse included patients’ decision to delay harvest or autologous stem cell transplant (ASCT) for the time being and return to the decision in the future. All interviews were audio-recorded and transcribed verbatim. Four investigators independently coded transcripts through inductive analysis to assess reasons for treatment decisions. Results: Of the 29 patients interviewed, 68% experienced treatment underuse: 21% self-identified as African American, 5% as Hispanic, 10% as mixed race, 57% as White, and 16% as Asian. There were no racial differences in reasons for underuse or delay. Themes relating to treatment underuse included: perceived pros and cons of treatment, including potential harm or lack thereof in delaying treatment; physician recommendations; and personal agency. Conclusion: Patients’ decision making, delays, and underuse of MM treatment are influenced by social, personal, medical, and contextual factors. Patients consider their relationship with their physician to be one of the most significant driving forces in their decisions and treatment plans.

## 1. Introduction

Multiple myeloma (MM) is a malignancy of plasma cells and is the second most common hematologic malignancy representing 1% of all cancers in the United States [1] with an estimated 35,000 new cases diagnosed in the year 2022 and approximately 12,000 deaths [2]. Nationally, African Americans are more likely than Whites to be diagnosed with MM (15.7 vs. 7.7 per 100,000) and to die of MM (7.5 vs. 4.2 per 100,000) [1]. Hispanics are also diagnosed with MM at younger ages than Whites (65 years vs. 70 years) [3] and have lower median survival times (2.4 years vs. 2.6 years, *p* = 0.006) [4].

Disparities in socioeconomic status and race exist in SCT survival rates [4,5,6,7,8]. There are disparities in the utilization of stem cell transplantation [4] as well as MM mortality [5]; however, transplant success and comparable survival from transplantation are equal when access to care is equal [4,5]. While we find these disparities in diagnosis and survival, studies continue to show that this disparity appears to be primarily related to socioeconomic status, access to care and new treatments, and inferior treatment use [9,10,11].

African American patients were less likely to undergo SCT compared to White patients (aOR = 0.49; 95% CI: 0.27–0.86) [2,12]. Underuse of maintenance therapy (aOR = 1.98; 95% CI 1.12–3.48) and interruptions in treatment were associated with race/ethnicity and insurance (aOR = 4.14; 95% CI: 1.78–9.74). Hispanic patients are shown to have less improved relative survival rates when compared to Non-Hispanic White and Non-Hispanic African American patients [13].

At the time of this study, the standard of care treatment for patients diagnosed with MM was induction therapy followed by high-dose melphalan and stem cell transplant (SCT) followed by ongoing maintenance therapy [14]; 33% of patients did not undergo SCT, and 10% did not receive maintenance therapy.

Understanding factors that influence patients’ treatment decision making and can affect rates of delayed or underused treatments may help identify areas to improve care or discussions about treatment recommendations. We undertook this study to better understand the factors associated with delayed and underused guideline adherent myeloma treatment.

## 2. Methods

Eligible patients were adults diagnosed with MM between 2010 and 2014 who were seen at Mount Sinai Hospital in New York City and who had experienced treatment underuse. Underuse was defined as either not receiving, not undergoing a stem cell transplant if eligible for transplant, and/or not receiving maintenance therapy if eligible, as these treatments reflected the standard of care for MM treatment during the study years [15].

A total of 1011 MM patients were screened, and based on the criteria above, 239 patients were eligible for interviews. All patients underwent induction unless there was a poor prognosis due to comorbidities, as documented in physician notes. No patients in the sample were classified as underuse of induction as we could not reach them. Patients classified as underuse of transplant were transplant-eligible and did not undergo autologous hematopoietic cell transplant after they had received induction therapy and had a partial response (PR) to induction therapy. Patients classified as underuse of maintenance therapy did not undergo maintenance therapy after they were either transplant-eligible and obtained a stable disease or a marginal, partial, or complete response following transplant or were transplant-ineligible and obtained a stable disease, marginal, partial, or complete response following induction therapy. Patients were ineligible if they completed all treatments or were classified as cognitively impaired by their physicians.

Eligible patients received a letter from their physician informing them about the study and giving them the opportunity to refuse to be contacted by the study team. After 5 business days, eligible patients were called to confirm eligibility, determine interest in participating, and obtain written informed consent if they were interested in participating. A total of 29 interviews were conducted between August 2019 and November 2020 in person or over the phone by experienced interviewers (NB or JL), audio-recorded and transcribed verbatim. Participants were asked about their understanding of their disease and treatments, factors regarding their treatment decisions, their treatment experience, and barriers to and facilitators of their care. Participants were asked to self-identify race, gender, marital status, nationality, level of education, religious preference, and health insurance status at the time of MM diagnosis and treatment. We aimed to recruit 25–30 patients to ensure theme saturation.

The study was approved by the Icahn School of Medicine at Mount Sinai Institutional Review Board.

## 3. Data Analysis

Analysis was completed through an inductive process. We (RC, JJL, NAB, RY) independently read through transcripts to identify themes and jointly reviewed transcripts to reach an agreement on code definitions and create a code key. We then independently coded each transcript and jointly reviewed them to resolve discrepancies in coding. We used Dedoose software to produce code reports.

## 4. Results

### 4.1. Patient Sample

We interviewed 29 patients with a MM diagnosis between 2010 and 2014 who experienced treatment underuse and delay, of whom 62% were female, 52% self-identified as White, 31% as African American: 10% as Hispanic, and 7% as Asian (Table 1). All patients received induction chemotherapy, 93% received stem cell harvest, 48% underwent transplantation, 45% received maintenance, and 7% chose to delay treatment (Table 1). The mean age of the cohort was 67 years (range 34–83) (Table 1). There was no racial difference between those who did and did not experience treatment.

### 4.2. Themes

Overall, we identified three main themes relating to treatment underuse or delay: (1) pro/con assessment; (2) physician recommendation; and (3) patient individual preferences for decision making. Patients who experienced underuse of SCT and those who experienced underuse of maintenance therapy both similarly assessed the pros and cons of treatments while making treatment decisions. There were no racial differences in reasons for underuse.

### 4.3. Assessment of Pros and Cons

Patients described considering perceived disadvantages of treatment relative to the possibility of remission or cure. Many explained that treatment side effects, their age or comorbidities, logistics of treatment, or their quality of life during treatment, including personal and familial social and emotional wellbeing, were weighed against the possible benefit of treatment (Table 2). One patient explained, “Why put my body at risk if [treatment] is not 100% proved” (PT 356).

Many added that being in remission after induction (and prior to undergoing SCT) was a significant influence on their decision to forgo further treatment. If they were currently feeling well and healthy, the risk of more treatment and potential pain was not worth the sacrifice of their current wellbeing. A patient stated, “Why are you going to subject your body and your life to this stem cell transplant, when you’re in remission?” (PT 360).

Patients described that if they were already in remission, seeking further treatment at the time felt unnecessary, and they felt supported in this decision knowing they had the option in the future, if they relapsed, to choose treatment. Further, patients acknowledged that treatment itself was not without its own risks, and these potential challenges, when measured against current remission, made decision making easier. One patient discussed:

“I basically said why would I need to use maintenance if … I’ve been in remission for a while with chemo and then stem cell…these are the risks… the more you use it, the less it works, so it’s like, well, why wouldn’t we save that for what I really need, if I do ever again”.(PT 1506)

Patients also emphasized weighing what they understood to be the respective benefits of therapy with the physical, emotional, and financial risks of continuing with treatment. These risks often included financial burdens or treatment logistics. A patient described the significant impact treatment had on their ability to function normally and that their worsening quality of life during treatment influenced their treatment decision making:

“I would come home and I had no energy to do anything at all. I still don’t have tons of energy, but I, I don’t stop. I don’t sit down. I don’t nap during the day. But if I want to go out, I don’t have to make excuses and say I’m tired, you know?...So why deal with [treatment] now when nothing is broke yet”.(PT 5)

Patients recalled emotional and psychological side effects of treatment and negative treatment effects on quality of life relative to current states of wellbeing when discussing deterrents to treatment. One patient described treatment and its effects as:

“poisons. even though they saved my life. I had like hyper moments and then I was, you know, would be so energized and then I would be exhausted. And it was just—took me through such ups and downs. I was terribly exhausted. I just felt like I was reeling from it, and was very, very, I guess depressed”.(PT 241)

Finally, patients described the difficulty of treatment logistics, including scheduling, location, and cost, as issues for themselves and their families. Patients cited the frequency and intensity of infusions and the distance and transportation to and from treatment centers as considerations in their decision making. A patient explained that following the intensity and involvement of infusions, “at least a month, two months in the hospital … building your immune system back up that you would be like in quarantine” (PT 408). Another described that logistically it was too difficult to reasonably get to the only hospital that could provide a transplant without transfusions.

“[They] knew it would cost money for me to go to Philadelphia, and it was like a money issue for me…it was going to be costly because of the traveling… That’s why I was more interested in doing chemo treatments here”.(PT 595)

### 4.4. Physician Recommendation

Patients discussed the recommendations their hematologists and specialists offered as primary influences on their treatment decisions. They valued their physicians’ honesty about treatment efficacy, side effects, and quality of life, and many noted they appreciated when their physicians acknowledged their mental and emotional health as well (Table 2). Adherence to physician recommendations depended on open communication and honesty with their healthcare teams as well as how well the recommendations aligned with patients’ individual preferences. Many felt confident in their physicians’ recommendations because they trusted their physicians. One patient recalled:

“I relied on [the doctor’s] advice because his advice was “you really don’t need to have a stem cell program,”… he said, “I don’t recommend it” and so from what I had right at the time that struck me as a reasonable way to go. I really didn’t probe a lot… It was sufficient for me and it was kind of consistent with my thoughts from what I had read that maybe I didn’t want to go that route”.(PT 1796)

Additionally, physicians’ recommendations were significant in patients’ decisions to not undergo SCT, especially when patients were in remission following induction therapy. Patients described being told transplant was still a possibility in the future if needed and feeling supported in their decision to forgo further treatment for now (Table 2). One patient described this experience: 

“[my doctor] said, I want you to do one more chemo treatment, which I did, and then when I saw him again, I guess it was after the treatment in October, he said, now you’re in complete remission. And I thought, I don’t want to go through that, I don’t want to go through that. And then I was so happy when [the doctor] said, no, he says, you’re in complete remission, he says, let’s wait, we got your cells. If we need them, we’ll use them”.(PT 1796)

### 4.5. Individual Preference for Decision Making

Patients also attributed their treatment decision making to their own personalities and values. They described individual priorities and characteristics as foundations for the types of decisions they made and emphasized how personal such decisions are (Table 2). A patient described how they preferred to make decisions that provided them safety nets and backup plans: “I harvested all of my stem cells. So that if I go—come out of remission, I can use my own stem cells to do my own transplant. All right, so yes, I’ve always been a contingency plan person” (PT 360). Another patient recalled previous medical experiences during which they faced similar risky or unfavorable odds and used those moments to support and guide their decision making. One patient said, “I’ve always been breaking the odds, and being the one who’s the stand alone so maybe [my doctor] wanted to use me in a medical sense to say maybe [I] can prove that [I] don’t need maintenance therapy, because I haven’t needed it four years later” (PT 1392).

Patients were able to identify prominent aspects of their personalities that were driving their decisions (Table 2). Another patient expressed a lifelong fear of needles and a stubborn personality as influences on their treatment decisions: “I tried to avoid, or it’s my nature, if I can fight something on my own, I’d rather do that… So being stubborn as all get out… I will use this time to rebuild myself as best I can” (PT 2332).

Finally, some patients explained how they sought out the experiences of other MM patients when making decisions. Specifically, myeloma patient groups and other cancer support groups were often utilized and helpful. Relationships among patients with MM and the advice they shared helped patients feel comfortable and more confident in their own personal decisions. One patient described “talking on the phone quite a bit” to fellow patients and recounted being advised, “if you can take a vacation from these drugs, take it” (PT 1392).

## 5. Discussion

We identified three primary themes contributing to MM treatment underuse: (1) pro/con assessment; (2) physician recommendations; and (3) patient’s own preferences for making decisions. Our work identifies the importance of patient-centered communication about MM treatment, specifically focusing on the emotional and psychological burdens of treatment and the effects of treatment on patients’ daily lives. Further, our results highlight that while there are similarities among influences on patients’ decisions, ultimately, treatment decisions are individual and affected by each patient’s individual priorities and perceptions about treatment advantages and disadvantages. Lastly, trust and communication in the patient–physician relationship are key and especially significant when a physician demonstrates an understanding of and empathy for a patient’s own health priorities.

Patients’ priorities that influence treatment decisions differ from those of their physicians [16,17,18,19,20,21]. Specifically, their personal life and personal and familial responsibilities are more central to their decision making [16,17]. Physicians take standards of care and comorbidities into consideration, in addition to the quality of life, when advising about treatment [16]. We similarly saw that patients value these considerations from their doctors and that patients personally consider the impact of treatment on their lives and on their families, finances, and mental wellbeing when making treatment decisions.

Patients’ decisions are often influenced by an understanding of current and potential wellbeing [16,22]. Specifically, MM patients made treatment decisions based on their state of health at the moment of the decision and were less influenced by potential health gains they could experience in the future with treatment. As a result, patients who were in remission after induction often did not feel that the possible benefit gained from additional treatment (e.g., SCT) outweighed the risks. Consequently, these patients in remission often would forgo treatment. Thus, recognizing that patients may weigh the pros and cons of treatment based on how they are currently feeling, as well as acknowledging differences in perceptions of risks and benefits, may help physicians better counsel MM patients about treatment options.

There are myriad differences between patient and physician priorities when it comes to shared decision making [16,17,20]. Tariman specifically concluded that while there were similarities in influential factors on decision making between older symptomatic MM patients and their physicians, the differences that do exist are significant for “decisional satisfaction” [16].

Patients in the Tariman study chose specific treatments based on previous experiences with health care, even over their physicians’ recommendations [16]. We similarly found that patients heavily considered their previous negative experiences with side effects, treatment, or the health care system as influences on their decision making. Specifically, we commonly saw that patients acknowledged their previous experiences in relation to their current status and treated decisions as not permanent but conditional on their physical and mental wellbeing.

When making treatment decisions, patients attributed more importance to the experiences they had previously had or the experiences of other MM patients than to potential outcomes of treatments about which they knew less [23]. We specifically found that MM patients sought out the successes of other MM patients to feel supported in their own goals and decisions. Patients found comfort and support when they were able to identify with other MM patients whose experiences contributed to their treatment decisions as well as personal wellbeing and a sense of support and community. Patients also experience support in their treatment, and their personal values and experiences are supported by their physicians [24,25]. It is important that such support in treatment decision making comes both from the patient’s medical team and their personal community. Counseling patients and providing comprehensive advice about treatment should include acknowledgment and understanding of the ways side effects and changes to the quality of life, including emotional, financial, and psychological tolls, contribute to patients’ decisions. Supporting these personal influences benefits patients’ understanding and acceptance of treatment.

Because of the trust and support patients had in their physicians, they were able to engage in discussions of future focused treatment of MM, though they had preferences to not follow guidelines at the present time. The long-term and incurable nature of MM have an influence on the perspective on and the decisions surrounding standards of care [26]. There is an understanding of and anticipation of the treatment process being long itself and being lifelong, and intentionally not following guidelines at present provides patients and physicians with choices in the future. It is significant that patients and physicians share trust and a willingness to counter guideline-recommended treatments and that such intentional decisions are frequent in this patient population. That many patients make such decisions, with the support of their physicians, highlights the importance of individual decision making in applying guidelines to individual patients’ treatment plans. The fact that a significant proportion of patients and their myeloma physicians were willing to work together and find common ground in sync with patient preferences and the changing state of the science is a testament to the scientific community’s ongoing learning and patient-centered approach to care.

In addition, the relationship patients share with their doctors and their doctor’s recommendations contributed significantly to their underuse of transplants. The relationship between patient and physician with respect to treatment recommendations and decisions includes the patients’ desire to participate in the treatment process and feel that their desires and preferences are being acknowledged and supported by their doctor [27]. A physician’s respect, honesty, and ability contribute to patients’ feelings of trust toward them [26,28,29,30]. Whitney et al. concluded that trust between a physician and patient was a combination of “internally validated trust… externally validated trust… [and] trust in relation to shared decision-making” [26]. Trust can be fostered between a physician and patient, especially through communication, time, demonstration of knowledge, and empathy, and improving trust can improve shared decision making [26,29]. Specifically, we found that not only is trust in and comfort with a physician integral to patients following the recommended treatment, but allowing for the possibility that treatment plans may change based on the quality of life and state of remission seemed to increase patient acceptance of physician recommendations.

Given the potential for patients with MM to remain in remission for a long time, patients and physicians were able to make treatment decisions together when they were reframed as “back-up plans”. A patient currently in remission and a physician recommending treatment often easily agreed that no additional treatment was needed until recurrence made further treatment necessary. This contingency planning provided patients with the ability to continue with their lives and make the decisions they wanted with the support and recommendation of their physicians, which in turn often led to improved feelings of being understood and supported by the medical team [31]. Providing current standard treatments as “back up” possibilities for the future as opposed to decisions that have to be made in the present would allow for less distress and a feeling of a united team between patient and physician.

## 6. Study Limitations

This study was conducted with participants recruited from one large tertiary urban hospital, and thus, our findings may not be generalizable to other MM populations. However, many of the themes we found have been borne out in other studies. Further, interviews were conducted with only English-speaking participants, so we were not able to assess language barriers as a potential factor in treatment underuse and decision making. In addition, interviews were conducted several years after the initial myeloma diagnosis and treatment and may be affected by patients’ recall and subsequent experiences. None of the patients experiencing induction underuse could be reached for an interview, and patients’ deaths may have resulted in some selection bias.

## 7. Conclusions

Our findings emphasize the influence of patients’ perceived risks and benefits of treatment, trust in their physician recommendations, and their individual practices about decision making on patients’ MM treatment choices. Reframing treatment as an option that continually exists for patients to choose from at future points, communicating consistently and openly about the effects of treatment on patients’ quality of life, and providing and focusing on achievable means of alleviating physical and emotional discomfort may help patients feel supported in their decisions and ease the difficulty of such decisions.

## Figures and Tables

**Table 1 cancers-15-02369-t001:** Sociodemographic characteristics of patient sample.

Variable	N	(%)
**Gender**		
Male	11	38%
Female	18	62%
**Age**		
18–45	1	3%
45–65	8	28%
65–80	19	66%
>80	1	3%
**Race**		
White	14	52%
African American	6	31%
Hispanic	3	10%
Asian	2	7%
**Treatment Received**		
Induction	29	100%
Stem Cell Harvest	27	93%
ASCT	14	48%
Maintenance	13	45%

**Table 2 cancers-15-02369-t002:** Themes related to treatment underuse or delay.

Theme	Definition	Quote
Assessment of treatment pro/cons	A consideration of relative risks and benefits of treatment, treatment side effects, or quality of life relative to effectiveness of treatment.	“It … took me through such ups and downs…” “why deal with [treatment] now when nothing is broke yet?”
Physician recommendation	Understanding of physician and/or medical team recommendation, specifically physician recommendation or support for delaying treatment while in remission.	“I relied on Dr. [redacted] advice because his advice was—you really, you really don’t, you really don’t need to have a stem cell program, he said, but, you know” in his own humorous and—style, and I mean this positively, he said, “You know, if you’d like me to inject you with yes themselves, we can arrange it.” But he said, “I don’t recommend it.”
Individual preference for decision making	Patients’ personal opinions, values, beliefs, experiences, or individual healthcare goals influence their decision making.	“I’ve been terrified of needles ever since I had a rabies shot at like age 6. So fewer needles. I try to avoid, or it’s my nature, if I can fight something on my own I’d rather do that… so being stubborn… I figured to a certain extent it’s what do I have to lose. I will use this time to rebuild myself as best I can.”

## Data Availability

No new data were created or analyzed in this study. Data sharing is not applicable to this article.

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
