# Peer review of "What Affects Treatment Underuse in Multiple Myeloma in the United States: A Qualitative Study"

_cancers, 2023, doi:10.3390/cancers15082369_

Round 1
Reviewer 1 Report
Congratulations on this very interesting paper.
In the following paragraph: "Underuse was defined as not being transplanted if eligible, and/or not receiving maintenance therapy. Underuse included patients’ decision to delay harvest or autologous stem cell transplant for the time being and return to the decision in the future". Underuse also includes transplant-ineligible patients with or without significant co-morbidities who were not properly informed about the availability of new, safe, feasible treatment modalities.
I suggest the following paper as a source or reference: Efficacy of first-line treatment options in transplant-ineligible multiple myeloma: A network meta-analysis. DOI: 10.1016/j.critrevonc.2021.103504
Reviewer 2 Report
- Define SCT when you first use it
-
Methods:
- It is not clear how the patients were identified exactly as being eligible for the study, what were the exact criteria and by which method were they identified ?
- What are the specific criteria used in this study for considering a patient eligible for ASCT? Age, comorbidities etc? How were these controlled for and recorded?
- In addition that title and abstract should make clear that only patients that are eligible for transplant are relevant to this study with regards to ASCT or maintenance. The title generally refers to myeloma treatment which also includes non-transplant eligible patients.
- The title should also should make clear that the study is conducted in the USA.
- Table 1 needs to be more analytical and include all relevant clinical information regarding patients (age, induction therapy, time of transplant, type of MM, ISS staging, comorbidities, how many had harvest but no transplant).
- The sample population is very small. I believe that a qualitative study on its own does not have much value with regards to the question addressed. I believe that a qualitative assessment could be part of an overall evaluation of factors that affect multiple myeloma under-treatment or underuse. A retrospective review of medical data is required to draw more robust conclusions.
- Overall the information that this study offers is very limited and could potentially be re-considered as a letter.
Reviewer 3 Report
GENERAL COMMENTS – MAJOR ISSUES
This is a study aims to investigate factors affecting treatment underuse, among multiple myeloma patients, by using a direct personal communication with the patients. Although the study design appears appropriate, it is surprising that only about one tenth of the eligible patients were approached and have provided their thoughts. It is a pity that so many, otherwise eligible patients were ignored, or ultimately for various reasons were not reached to be enrolled in this analysis. I would consider as an acceptable percentage of not achieving a contact with the eligible patients between 20 and 30%, or even 40%, but not 88%. The authors need to explain why they opted to go ahead with the analysis of only 29 patients’ feedback and have not insisted in substantially increasing this number, which would render their study more representative. Further, the authors have found that several of their patients had underused the maintenance phase of anti-myeloma treatment. However, it is not easily understood why a patient or a doctor would decide to defer maintenance treatment, which is usually taken easily, does not require frequent hospital visits and is rather well-tolerated. In other words, I could understand the underuse of the consolidation phase / High-dose therapy and Auto-SCT in a patient who might be reluctant and might decide not to go ahead, but for the maintenance treatment the major, if not the only reason for its underuse, might be the absence of health insurance or of insurance-related lack of financial support of this type of treatment. This reason needs to be clearly defined, investigated and analyzed by the authors. Moreover, the authors should have analyzed and provide information on the role of patient’s psychological support applied by expert/psychologists and not by the treating physicians. Psychological support could help patients in doubt, to reach the appropriate and recommended option. Has this parameter been evaluated at all in this study? Finally, the authors could have presented data on the impact of patients’ personal perceptions and information obtained through sources other than the treating physicians (second expert’s opinion, Influencers through the Internet, etc). Has this parameter been investigated during the conversation with the patients?
On top of the above, several important issues have not been investigated and are not reported. First, nothing is mentioned about the impact of patient’s age, race and gender in treatment underuse.
Second, nothing is mentioned about the role of patient’s family or of Myeloma/Cancer patient groups or even of Specific Multiple Myeloma Foundations. If a patient is at risk to make erroneous options, which might unwillingly affect his/her life, these supportive back-ups might influence the restoration of the appropriate decision. The authors should comment on these and other factors.
Third, the authors should explain why they have considered treatment underuse only the consolidation and maintenance phase of treatment (Page 2, Methods, lines 15-17). Couldn’t they have identified discrepancies regarding induction treatment among the 1011 patients? They could for example consider underuse the application of two, instead of three drugs as induction treatment or the application of fewer than 4 or 6 cycles of induction treatment. Have all of their patients received the indicated induction treatment? This needs to be emphasized.
Fourth, as previously mentioned, the authors should also explain why they had planned to recruit only 25-30 of the 239 eligible patients (i.e. about 10-12%) (Page 2, Methods, Lines 29-30) and had not opted to obtain a broader recruitment sample, to achieve more representative number of patients and clearer messages.
Fifth, it is not easily understandable in what way a physician recommendation might result in underuse of treatment in a myeloma patient (Page 4, paragraph 4.4). Does this mean that some physicians might advise their patients against the existing guidelines for treatment of myeloma? What kind of physicians provide scientific advice to myeloma patients? Are these physicians GPs or family Doctors or are they Hematologists-specialists? The reader of the manuscript might think that family (non-expert) doctors might exert high level of confidence and respect to their patients and advise them erroneously, on a different way than expert hematologist might recommend. Is this the message? If yes, this needs to be further clarified.
Sixth, Table 2 should include as additional themes the underuse of Induction treatment in terms of appropriate drug composition and minimal duration to achieve at least a VGPR. Regarding the second theme entitled as “Physician Recommendation” the authors need to make clear what kind of physicians are referred. Are they referred to treating physicians/specialists or to family doctors/GPs, who might have stronger relations with the patients?
Reviewer 4 Report
This paper purposed to better understand the factors associated with delayed and underused guideline adherent myeloma treatment. I do have some comments as listed below in the order noted.
Comment 1: What is the novelty of this study although several “poor prognostic factors with delayed and underused guideline adherent myeloma treatment” prediction models have been proposed earlier?
Comment 2: In Introduction, the authors mentioned that disparities in socioeconomic status and race exist in SCT survival rates. The authors should cite the references
Maignan K, et al., Blood Cancer J 2022;12(4):65
https://www.nature.com/articles/s41408-022-00665-x;
Xu J et al., Front Oncol 2022;12:941714;
https://doi.org/10.3389/fonc.2022.941714
Chamoun K et al., Cancers (Basel) 2021;13(4):590,
DOI: 10.3390/cancers13040590
which were to identify the disparities in socioeconomic status and race exist in SCT survival rates among a patient with MM.
Comment 3: The quality of the data set is very important, especially in an observational qualitative study. For this reason, please specifically clarify the inclusion criteria and exclusion criteria of sample collection in the Methods section.
Comment 4: Please provide the Strengths or Significances of the study in the Discussion section.
Reviewer 5 Report
Please make this sentence more clear- Disparities in socioeconomic status and race exist in SCT survival rates [4-8], although studies continue to show that this disparity appears to be primarily related to socioeconomic status, access to care and new treatments, and inferior treatment use [9-11].
Write full form of SCT , aOR when you use it first time in paper.
In your abstract you have mentioned 28 patient s but in body of paper it is 29. Please clarify.
It is an interesting small scale qualitative observational study to assess patient's belief regarding their treatment choices. I am not sure by reading this study there will be any major change in physician behavior which will lead to change in treatment outcome. There was no information if not going through entire recommended treatment, there was worse outcome for patients.
Round 2
Reviewer 2 Report
No further comments, accept in present form as a qualititative initial study that addresses this specific issue.